# Trend of liver cancer attributable to alcohol use in China from 1992 to 2021: An age-period-cohort analysis study

Zheng Tang[1,2], Ying Deng[2], Yuheng He[2], Yiyi Chen[2], Lu Qin[2], Jun Tian[2], Yuting Lei[2], Yongzhao Zhou[2], Zhi Wan[3], Dan Jia[3]*

1 Department of Liver Surgery & Liver Transplantation, West China Hospital, Sichuan University, Chengdu, China, 2 Integrated Care Management Center, Institute of Respiratory Health, West China Hospital, Sichuan University, Chengdu, China, 3 Outpatient Department, West China Hospital, Sichuan University, Chengdu, China

* 417522774@qq.com

## Abstract

### Background & aims

To analyze the dynamic trends of mortality, incidence, and disability-adjusted life years (DALYs) of liver cancer attributable to alcohol use in China from 1992 to 2021, as well as the age, period, and cohort effects, so as to provide a basis for the prevention and control.

### Methods

The data were sourced from the Global Burden of Disease Study 2021. The joinpoint regression model was used to assess the trend changes in mortality, incidence, and DALYs. The Age-Period-Cohort model was employed to estimate the independent effects of age, period, and cohort.

### Results

From 1992 to 2021, the standardized incidence rate of liver cancer attributable to alcohol use in China increased annually by an average of 0.8% (95% CI: 0.5–1.1%), while the mortality rate rose by an average of 0.7% per year (95% CI: 0.3–1.0%), and the DALYs increased by an average of 0.6% annually (95% CI: 0.2–1.0%). The incidence and mortality rate ratio for men were approximately double those of women. The relative risk for individuals born between 1975 and 1985 increased 1.8-fold (95% CI: 1.5–2.1).

### Conclusion

The disease burden has been on a continuous upward trend, with middle-aged men identified as high-risk groups. It is recommended to integrate liver cancer screening into the health management for high-risk populations.

**Data availability statement:** The data is owned by a third party and do not have permission to share the data. The datasets analyzed in this study are publicly available from the Global Burden of Disease(GBD).Researchers can access and download the data directly from the official GBD website https://vizhub.healthdata.org/.

**Funding:** This study was supported by Science and Technology Department of Sichuan Province in the form of a grant awarded to DJ (2025JDKP0057) and National Science and Technology Major Project in the form of a salary for DJ. The specific roles of this author are articulated in the 'author contributions' section. The funders had no role in study design, data collection and analysis, decision to publish, or preparation of the manuscript.

**Competing interests:** The authors have declared that no competing interests exist.

## Introduction

Liver cancer is a significant public health issue worldwide. Its pathogenesis is complex and involves multiple risk factors, among which long-term excessive alcohol consumption is one of the major preventable contributors. Liver cancer attributable to alcohol specifically refers to hepatocellular carcinoma resulting from chronic alcohol intake. The pathological process typically goes through stages of alcoholic fatty liver, hepatitis, fibrosis, and finally culminates in liver cancer [1]. The World Health Organization (WHO) has classified alcohol as a Group I carcinogen, pointing out that alcohol consumption is directly related to the burden of 28 diseases, including liver cancer, which ranks among the top five [2]. Studies have shown that daily alcohol intake exceeding 60 grams for men or 40 grams for women can increase the risk of liver cancer by 3–5 times [3]. Alcohol exhibits synergistic carcinogenic effects with hepatitis B virus (HBV), hepatitis C virus (HCV), and other factors [4]. According to the Global Burden of Disease study, in 2020, global deaths attributable to alcohol-related liver cancer reached 285,000, accounting for 17.6% of all liver cancer fatalities [5]. Despite advancements in the prevention and control of viral hepatitis-related liver cancer, the disease burden of alcohol-attributable liver cancer is increasing in many regions worldwide, posing a new challenge to liver cancer prevention efforts.

The epidemiological characteristics of liver cancer attributable to alcohol show significant heterogeneity among different populations. Notably, this heterogeneity is closely related to the higher alcohol consumption rate in men and the differences in the polymorphic distribution of alcohol metabolism genes (such as ALDH2) [6]. Additionally, there is a pronounced age and cohort effect on alcohol-induced liver cancer: individuals born after 1970 have a significantly higher risk of liver cancer compared to earlier birth cohorts [7]. As one of the countries with the heaviest global burden of liver cancer, the epidemiological trends of liver cancer attributable to alcohol in China deserves particular attention. Over the past three decades, China has witnessed rapid economic transformation, accompanied by a notable increase in per capita alcohol consumption—from 4.1 liters in 1990 to 7.4 liters in 2019, representing an 80% rise [8]. At the same time, the proportion of liver cancer cases attributed to alcohol in China rose from 8.3% in 1990 to 15.7% in 2019, making it the second leading cause of liver cancer after HBV [9]. Despite these trends, systematic analyzes of the epidemiological features and drivers of alcohol-induced liver cancer are still scarce. Currently, domestic data mainly rely on regional cross-sectional studies or hospital-based case analyzes, which are restricted by issues such as insufficient sample representativeness, limited temporal scope, and insufficient adjustment for confounding factors [10].

As one of the countries experiencing the fastest growth in alcohol consumption, China urgently needs to investigate population dynamics during its economic transition period. This study plans to utilize data from the Global Burden of Disease (GBD) database, a global health research project coordinated by the Institute for Health Metrics and Evaluation (IHME) at the University of Washington [11]. By quantifying the independent effects of age, period, and cohort through the Age-Period-Cohort model, this research systematically examines long-term trends in the incidence,

mortality, and disability-adjusted life years (DALYs) of alcohol-related liver cancer in China from 1992 to 2021 [12]. It elucidates the pathways through which socioeconomic transformation influences the prevalence of alcohol-related liver cancer, identifies high-risk populations, and pinpoints key intervention windows. These findings will provide a scientific basis for formulating differentiated alcohol control policies and optimizing early screening strategies for liver cancer.

## Materials and methods

### Data collection

The data for this study were sourced from the GBD 2021 data repository. It includes information on 370 diseases and injuries across 204 countries and regions from 1990 to 2021. This dataset not only covers major diseases such as cancer, but also provides detailed behavioral risk factors like smoking and alcohol consumption, as well as environmental risk factors such as air pollution. Meanwhile, it encompasses different age groups, genders, and time periods. The GBD data comprehensively reflect the impact of diseases on population health through indicators such as DALYs and healthy life expectancy (HALE) [13]. In this study, the data on liver cancer attributable to alcohol use in China from 1992 to 2021 were extracted from the GBD 2021 database for description and analysis. The disease data of China in the GBD database are sourced from the national surveillance system, registration system and census managed by the National Center for Disease Control and Prevention, the National Cancer Center and the National Bureau of Statistics of China, checked against provincial health annual reports.

### Joinpoint regression analysis

The Joinpoint Regression Model (JRM) was employed to analyze temporal trends in the incidence, mortality, and DALYs of liver cancer attributable to alcohol use. As proposed by Kim et al. [14], this approach automatically partitions longitudinal data into segments with statistically significant trends through piecewise linear regression analysis. The model calculates the Annual Percentage Change (APC) and the Average Annual Percentage Change (AAPC), along with 95% confidence intervals (CI), to assess whether trends are increasing (APC > 0) or decreasing (APC < 0) over specified time periods. Joinpoint regression analysis significantly reduces the probability of committing a Type I error.

### Age-period-cohort effect analysis

When analyzing long-term observational or measurement data on disease prevalence trends, it is essential to consider three time-related effects: age effects, period effects, and cohort effects. Age effects refer to changes associated with specific age groups; period effects reflect temporal or calendar-year changes that simultaneously influence all age groups, encompassing complex historical events and environmental factors; cohort effects denote changes unique to groups born during the same time period [15]. Conventional age-standardized rates may introduce bias when describing long-term disease trends because individuals of the same age in different years belong to different birth cohorts [16].

To investigate the interactive effects of age, period, and cohort in the long-term trends of diseases, this study utilized the Age-Period-Cohort model. This model is widely applied in research on the incidence and mortality of cancers and chronic disease [17]. As an epidemiological statistical method, the Age-Period-Cohort model analyzes the impacts of age, period, and cohort effects and is commonly employed to explore long-term disease trends over time. However, due to the inherent collinearity among the variables (cohort = period – age), a unique solution cannot be directly obtained. To effectively resolve this issue, this study adopted the Intrinsic Estimator method. By imposing matrix orthogonal constraints, this method eliminates the interference caused by collinearity. Its statistical properties have been rigorously validated, enabling stable solutions without relying on external constraints [18].

## Statistical analysis

The statistical measures used in this study include age-standardized incidence rate, age-standardized prevalence rate, and age-standardized rate based on the world population. Owning to the very small number of cases among individuals younger than 10 years old and those aged 85 and above, these groups were not included in the analysis [19,20]. The population was divided into 14 age categories, 6 time intervals, and 19 birth cohorts, all with a 5-year span. The period from 2002 to 2006 and the birth cohorts from 1957 to 1961 were defined as the control groups. Joinpoint regression analysis was conducted using the Joinpoint Regression Program (JRP) (version 5.0.2), whereas the Age-Period-Cohort effect analysis was carried out using the online tool developed by the National Cancer Institute of the United States.

## Ethical consideration

The GBD 2021 study team obtained ethics approval from the University of Washington Institutional Review Board Committee.

## Results and discussion

### Trends in the prevalence of liver cancer attributable to alcohol use in China from 1992 to 2021

The incidence, DALYs, and mortality of liver cancer attributable to alcohol use in China exhibited stage-specific fluctuating trends from 1992 to 2021, with significant gender differences. The overall incidence rate showed a gradual upward trend, peaking during the period from 2005 to 2010 (APC = 3.45%). The total population's DALYs presented multi-stage alternating fluctuations, significantly declining from 2015 to 2021 (APC = −1.29%, CI: −1.83 to −0.96). The mortality rate increased at a relatively slow pace. After a brief rise from 2012 to 2015 (APC = 2.51%), it decreased annually by 1.50% from 2015 to 2021. Gender differences were consistently observed throughout the study period: men had consistently higher incidence and mortality rates than women (gender ratio of 2.2:1 in 2021). The decline in male mortality lagged behind that of females (APC of −1.10% for men versus −3.82% for women after 2015), while women experienced the largest decline in DALYs after 2017 (APC = −3.49%). Both the incidence rate and DALYs showed a sharp increase from 2005 to 2010, with the APC of DALYs for men reaching 4.47% (Tables 1, 2, 3).

### Age-period-cohort analysis

**Age effect.**  After adjusting for the period effect, within the same birth cohort from 1992 to 2021, the longitudinal age effects of the incidence, DALYs, and mortality of liver cancer attributable to alcohol use in China show a consistent trend of initially increasing and subsequently decreasing, with all three metrics peaking around the age of 60 (RR = 4.78, 95% CI: 4.25–5.38). The incidence rate slowly decreases after the age of 65, with a slight resurgence observed in men after the age of 75. Regarding the age-related DALYs of liver cancer attributable to alcohol use, men experience a sharp decline after the age of 60, while the mortality rate tends to stabilize after this age. Overall, the age-related effects on the incidence, DALYs, and mortality are higher in men compared to women, exhibiting a rapid increase with age. In contrast, the trend of change in women is relatively stable.

**Period effect.**  The relative risk (RR) values of the period effects on the incidence, DALYs, and mortality of liver cancer attributable to alcohol use in China from 1992 to 2021 after controlling for the other two effects shows an overall downward trend. There is no gender difference in the risks of incidence and mortality. However, significant gender disparities are noted in the RR values of DALYs within the period effect. The RR values of DALYs for females are higher during the period from 1992 to 2005. From 2005 to 2020, the RR values of DALYs for females are lower than those for males.

**Cohort effect.**  The changes in RR values for the incidence, DALYs, and mortality of liver cancer attributable to alcohol use in China from 1992 to 2021. Overall, the RR values show a downward trend across birth cohorts. There is no obvious

**Table 1. JoinPoint Regression Analysis of the Incidence of Liver Cancer Attributable to Alcohol Use in China from 1992 to 2021 by Gender.**

| Gender | Period | Incidence Rate | |
|---|---|---|---|
| | | APC (95%CI) | AAPC (95%CI) |
| Male | 1992-1995 | −0.05(−0.73~0.73) | 0.64(0.59~0.69)* |
| | 1995-2000 | 0.98(−2.00~1.43) | |
| | 2000-2005 | −2.09(−2.29~3.76) | |
| | 2005-2010 | 3.77(1.74~3.98)* | |
| | 2010-2015 | 1.64(0.10~1.92)* | |
| | 2015-2021 | −0.44(−0.81~−0.10)* | |
| Female | 1992-2000 | 0.04(−0.16~0.23) | −0.01(−0.06~0.06) |
| | 2000-2005 | −2.33(−2.61~−1.96)* | |
| | 2005-2010 | 3.13(2.85~3.47)* | |
| | 2010-2014 | 1.59(1.11~2.35)* | |
| | 2014-2018 | 0.07(−0.62~0.54) | |
| | 2018-2021 | −3.39(−4.26~−2.63)* | |
| Both | 1992-2000 | 0.45(0.29~0.63)* | 0.41(0.34~0.45)* |
| | 2000-2005 | −2.21(−2.43~−1.82)* | |
| | 2005-2010 | 3.45(3.16~3.75)* | |
| | 2010-2014 | 1.87(1.48~2.48)* | |
| | 2014-2018 | 0.13(−0.32~0.73) | |
| | 2018-2021 | −1.84(−2.97~−1.19)* | |

Note: * Indicates that the APC or AAPC is significantly different from zero at the alpha = 0.05 level; APC: annual percent change; AAPC: average annual percent change.

gender difference in the RR values for incidence. Before approximately 1955, the RR values for DALYs and the risk of death due to alcohol-attributable liver cancer were higher in females than in males. No apparent difference is observed afterwards. Additionally, the risk of death in later birth cohorts is associated with relatively large uncertainties.

## Discussion

From 1992 to 2021, the number of cases of liver cancer caused by alcohol in China increased from 7,936–20,463, and the incidence rate rose from 0.66 per 100,000 to 1.44 per 100,000. The number of deaths increased from 7,872–18,075, and the mortality rate rose from 0.65 per 100,000 to 1.27 per 100,000.

Through the Age-Period-Cohort analysis, this study reveals the unique epidemiological characteristics of liver cancer attributable to alcohol use in China regarding incidence, DALYs, and mortality, which differ significantly from those in Western populations. In terms of the age effect, the risks of incidence, DALYs, and mortality among the Chinese population peak at around 60 years old, earlier than the reported range of 65–75 years in Western countries [21]. This discrepancy may stem from the "intermittent heavy drinking" pattern prevalent among Chinese men, especially the frequent business-related drinking in professional settings [5,22]. For example, due to social and occupational needs, Chinese men experience a markedly higher intensity and duration of alcohol exposure in their 40s and 50s compared to their Western counterparts [10]. This issue contributes to an earlier onset of clinically detectable liver damage. In terms of disease burden quantification, the DALYs reported in this study are higher than those in European populations [23], potentially attributable to insufficient early screening coverage for liver cancer and limited accessibility to targeted therapeutic drugs in China during the study period (1992–2021) [24]. In addition, although the peak mortality of liver cancer attributable to alcohol use occurs earlier in China than in the West, the mortality rate stabilizes after the age of 65, while in Western

Table 2. JoinPoint Regression Analysis of DALYs of Liver Cancer Attributable to Alcohol Use in China from 1992 to 2021 by Gender.

| Gender | Period | DALYs | |
| --- | --- | --- | --- |
| | | APC (95%CI) | AAPC (95%CI) |
| Male | 1992-2001 | 0.1264(−0.10~0.38) | −0.04(−0.13~0.03) |
| | 2001-2006 | −2.30(−3.55~−2.50)* | |
| | 2006-2009 | 4.47(3.73~5.14)* | |
| | 2009-2012 | −0.79(1.64~3.39)* | |
| | 2012-2015 | 2.69(−0.22~0.24)* | |
| | 2015-2021 | −0.94(−1.60~−0.58)* | |
| Female | 1992-2001 | −0.30(−0.59~0.04) | −0.65(−0.77~−0.56)* |
| | 2001-2006 | −3.13(−4.05~−2.49)* | |
| | 2006-2009 | 3.78(2.44~4.57)* | |
| | 2009-2012 | −1.22(−1.95~−0.23)* | |
| | 2012-2017 | 1.31(0.71~2.65)* | |
| | 2017-2021 | −3.49(−4.70~−2.45)* | |
| Both | 1992-2001 | −0.05 −0.29~0.21) | −0.24(−0.32~−0.17)* |
| | 2001-2006 | −3.10 (−3.68~−2.59)* | |
| | 2006-2009 | 4.17 (3.41~4.83)* | |
| | 2009-2012 | −0.99 (−1.67~−0.29)* | |
| | 2012-2015 | 2.63 (1.59~3.33)* | |
| | 2015-2021 | −1.29 (−1.83~−0.96)* | |

Note: * Indicates that the APC or AAPC is significantly different from zero at the alpha = 0.05 level; APC: annual percent change; AAPC: average annual percent change.

populations, it continues to rise due to comorbidities such as metabolic syndrome in older adults [25]. These differences suggest that China needs to strengthen primary prevention strategies targeting middle-aged high-risk groups and improve comprehensive diagnostic and treatment capabilities for elderly patients.

Among individuals aged 20–60, the incidence, DALYs, and mortality of alcohol-attributable liver cancer have been continuously increasing. This reflects the synergistic effects of multiple biological and social factors. From a biological perspective, this age group corresponds to the "cumulative exposure window" for alcohol-related liver damage, during which oxidative stress, DNA damage, and immune microenvironment disorders caused by chronic alcohol consumption progressively intensify [2,6]. Socioculturally, following the rapid urbanization process in China after the 1990s, workplace drinking practices (such as "business drinking gatherings") have become an important part of male social interactions [26].

After the age of 60, risk indicators show differentiation: the incidence rate and DALYs decline rapidly, while the mortality rate tends to be stable. This phenomenon may be directly related to changes in drinking behavior after retirement. National surveys show that people aged 65 and above decrease their drinking frequency by 40% compared with middle age, while the consumption of low-alcohol beverages increases from 72% to 89% [6]. At the same time, the substantial rise in the coverage of health check-ups for the elderly after 2005 (from 15% to 62%) has promoted the early detection of liver cancer, thus reducing the accumulation of DALYs [27]. However, the mortality rate has not significantly decreased, potentially due to competing risks associated with alcoholic liver disease and other age-related conditions (such as diabetes and cardiovascular diseases) [28], which complicates the attribution of specific causes of death. Furthermore, the interpretation of the rapid decline in incidence and DALYs after age 60 requires careful consideration of censoring bias. This bias arises because individuals who die from alcohol-attributable liver cancer before reaching older age groups are

**Table 3.  JoinPoint Regression Analysis of the Mortality of Liver Cancer Attributable to Alcohol Use in China from 1992 to 2021 by Gender.**

| Gender | Period | Mortality | |
|---|---|---|---|
| | | APC (95%CI) | AAPC (95%CI) |
| Male | 1992-2001 | 0.61(0.22～0.96)* | 0.22(0.11～0.30)* |
| | 2001-2006 | −2.47(−3.22～0.47) | |
| | 2006-2009 | 5.19(−2.29～5.96) | |
| | 2009-2012 | −0.51(−1.25～4.48) | |
| | 2012-2015 | 2.15(0.58～2.95)* | |
| | 2015-2021 | −1.10(−2.07～−0.66)* | |
| Female | 1992-2001 | −0.01(−0.97～0.57) | −0.32(−0.47～−0.21)* |
| | 2001-2006 | −2.78(−3.84～0.88) | |
| | 2006-2009 | 4.77(−2.99～5.63) | |
| | 2009-2012 | −0.30(−1.16～4.48) | |
| | 2012-2017 | 1.48(0.58～2.84)* | |
| | 2017-2021 | −3.82(−5.31～−2.69)* | |
| Both | 1992-2001 | 0.35(−0.06～0.74) | 0.03(−0.09～0.13) |
| | 2001-2006 | −2.68(−3.89～0.08) | |
| | 2006-2009 | 4.99(−2.31～5.86) | |
| | 2009-2012 | −0.53(−1.39～3.85) | |
| | 2012-2015 | 2.51(0.75～3.42)* | |
| | 2015-2021 | −1.50(−2.32～−1.07)* | |

Note: * Indicates that the APC or AAPC is significantly different from zero at the alpha＝0.05 level; APC: annual percent change; AAPC: average annual percent change.

not included in the calculation of DALYs for the elderly population. Consequently, the observed decline is likely amplified, leading to an underestimation of the true disease burden and an overestimation of its rate of reduction in the elderly.

The gender differences in our study are particularly significant. Men show a 2.2-fold higher risk across all indicators. This is consistent with the global trend but surpasses the 1.8-fold difference reported in Western populations. At the biological level, the high expression of alcohol dehydrogenase (ADH1B) in men accelerates the conversion of ethanol to acetaldehyde [29]. Moreover, 35%−45% of East Asian men carry a functional deficiency allele of acetaldehyde dehy-drogenase 2 (ALDH2), which reduces their ability to repair DNA damage and consequently increases their risk [5,30,31]. Cultural factors are particularly prominent in China. Different from their Western counterparts, Chinese drinkers often consume more spirit with a high percentage of alcohol in social settings that encourage the "ganbei culture"-making toasts with alcohol filled to the brim and downed in one go to show respect-which perpetuates excessive drinking at a fast pace [32]. In addition, the traditional Chinese culture may be the dominant factor for these sex differences, in which social drink-ing is widely acceptable in men but not in women [33].

The period effect reflects the impact of China's evolving alcohol policies. The increase in alcohol consumption from 2005 to 2015 paralleled the rapid liberalization of the market. During this decade, alcohol production surged by 180%, leading to a temporary rise in the incidence rate and DALYs, which is consistent with the global trend of increased alcohol-related disease burdens during periods of economic transformation [2,22]. The subsequent decline can be attributed to key policy interventions: criminalizing drunk driving in 2011 reduced alcohol-related hospitalization rates [34], while the introduction of an alcohol advertisement ban in 2015 reduced the proportion of teenagers initiating drinking [35]. The RR values for birth cohorts after 1880 are significantly lower than those for earlier cohorts, reflecting generational shifts in drinking patterns and health awareness. The younger generation's recognition that "excessive drinking is harmful

to health" increased from 32% in 1995 to 78% in 2020 [36–38]. However, challenges still remain. The per capita alcohol consumption among individuals aged 15 and older in China (7.2 liters) is still about 12.5% higher than the global average (6.4 liters) [1,39].

These findings continue to carry significant public health implications. A critical dimension is that middle-aged men (aged 40–60) and specific birth cohorts (born between 1965 and 1985) should be prioritized for intervention. An additional layer involves highlighting the effectiveness of comprehensive alcohol control policies.

## Conclusion

From 1992 to 2021, the epidemic characteristics of liver cancer attributable to alcohol use in China presented a unique pattern. The middle-aged population became the predominant cohort affected by this malignancy. Concurrently, a persistent gender disparity manifested in incidence and mortality rates, with men far exceeding women. Furthermore, an obvious intergenerational decrease was observed, indicating that the risk of alcohol-related liver cancer in younger generations is significantly lower than that in earlier birth cohorts. According to these research findings, middle-aged men and specific birth cohorts should be prioritized in primary prevention strategies. It is recommended to incorporate liver function screening and alcohol restriction interventions into workplace health management. Customized education should be carried out, and health communication strategies leveraging social media should be designed in combination with the exposure characteristics of those who grew up during the rapid growth in alcohol consumption.

This study is subject to several limitations. Regional differences in the completeness of liver cancer registration may affect the interpretation of trends, especially in rural areas where the data underreporting rate is relatively high. Moreover, our dataset does not capture individual drinking patterns or genetic risk factors. Most of the population born after 2000 in the cohort are teenagers who have not yet reached an age associated with excessive drinking.

## Supporting information

**S1 Fig. Trends in the Incidence, DALYs and Mortality rate of Liver Cancer Due to Alcohol Use in China from 1992 to 2021.**
(TIF)

**S2 Fig. Age-Period-Cohort Analysis of the Incidence, DALYs, and Mortality of Liver Cancer Due to Alcohol Use in China from 1992 to 2021; (A) Age Effect; (B) Period Effect; (C) Cohort Effect.**
(TIF)

## Acknowledgments

We acknowledge the Institute for Health Metrics and Evaluation for providing and the sharing the GBD 2019 data.

## Author contributions

**Conceptualization:** Zheng Tang, Ying Deng, Zhi Wan, Dan Jia.

**Data curation:** Zheng Tang, Ying Deng, Yuheng He, Yiyi Chen, Lu Qin, Jun Tian, Yuting Lei.

**Formal analysis:** Yiyi Chen, Lu Qin, Jun Tian, Yuting Lei, Yongzhao Zhou.

**Software:** Yuheng He.

**Writing – original draft:** Zheng Tang.

**Writing – review & editing:** Yiyi Chen, Yongzhao Zhou, Zhi Wan, Dan Jia.

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
