## [Decision Letter · Decision Letter 0]

18 Nov 2025

Dear Dr. Jia,

Thank you for submitting your manuscript to PLOS ONE. After careful consideration, we feel that it has merit but does not fully meet PLOS ONE’s publication criteria as it currently stands. Therefore, we invite you to submit a revised version of the manuscript that addresses the points raised during the review process.

Please submit your revised manuscript by Jan 02 2026 11:59PM. If you will need more time than this to complete your revisions, please reply to this message or contact the journal office at plosone@plos.org . A rebuttal letter that responds to each point raised by the academic editor and reviewer(s). You should upload this letter as a separate file labeled 'Response to Reviewers'.A marked-up copy of your manuscript that highlights changes made to the original version. You should upload this as a separate file labeled 'Revised Manuscript with Track Changes'.An unmarked version of your revised paper without tracked changes. You should upload this as a separate file labeled 'Manuscript'.

We look forward to receiving your revised manuscript.

Kind regards,

Ricardas Radisauskas

Academic Editor

PLOS ONE

**Journal Requirements:**

“National Science and Technology Major Project (2024ZD0523904)1·3·5 projects for Artificial Intelligence, West China Hospital, Sichuan University (ZYAI24015)Please state what role the funders took in the study.”

If the funders had no role, please state: "The funders had no role in study design, data collection and analysis, decision to publish, or preparation of the manuscript."

4. Please note that funding information should not appear in any section or other areas of your manuscript. We will only publish funding information present in the Funding Statement section of the online submission form. Please remove any funding-related text from the manuscript.

6. We note that there is identifying data in the Supporting Information file <IHME-GBD_2021_DATA-025a4d71-1.zip >. Due to the inclusion of these potentially identifying data, we have removed this file from your file inventory. Prior to sharing human research participant data, authors should consult with an ethics committee to ensure data are shared in accordance with participant consent and all applicable local laws.

-Location data

Additional Editor Comments (if provided):

Reviewers' comments:

Reviewer's Responses to Questions

**Comments to the Author**

1. Is the manuscript technically sound, and do the data support the conclusions?

Reviewer #1: Yes

Reviewer #2: Yes

2. Has the statistical analysis been performed appropriately and rigorously?

Reviewer #1: I Don't Know

Reviewer #2: Yes

3. Have the authors made all data underlying the findings in their manuscript fully available?

Reviewer #1: Yes

Reviewer #2: Yes

4. Is the manuscript presented in an intelligible fashion and written in standard English?

Reviewer #1: Yes

Reviewer #2: Yes

Reviewer #1: The current study investigated the burden of liver cancer attributable to alcohol use in China between 1992 and 2021 using the Global Burden of Disease (GDB) database. It found a steady increase in incidence, mortality, and disability-adjusted like years (DALYs), with middle-aged men identified as the highest-risk group. The results highlight generational and policy-related influences and underscore the need for targeted screening and prevention strategies in China. Strengths of the study include its timely and relevant aims of alcohol consumption in China and the use of a large and representative dataset (i.e., GBD data). However, several limitations reduce enthusiasm for the current project.

1) There is a lack of detail on how data quality was assessed. Explicit information on how GDB data for china were validated should be added to the manuscript.

2) Were multiple comparisons controlled for in the analyse? Is there the potential for inflated Type-I error? Either this statistical control should be used, or an explanation on why it is not needed should be provided.

3) The project determined that the incidence of DALYs declines after age 65. However, could this reflect mortality levels (i.e., individuals with liver disease died at earlier ages) rather than a true reduction in DALYs? In this case, I believe this data would be censored and may need to be accounted for. This should be discussed as a potential limitation or controlled for.

4) The figures are difficult to see as they are very blurry. Please provide more appropriate figures for interpretation.

a. Further, it appears that no confidence intervals are included on Figure 1. It is difficulty to see.

Reviewer #2: To my opinion, the manuscript is well-written and demonstrates the interesting results for researchers working in similar area as authors. But still I have some recommendations how to improve the quality of the manuscript. To my opinion results presented in Figure 1 partially duplicates the results from Tables 1-3. So I recommend remove from the Figure 1 results of Joinpoint analysis which are already presented in the mentioned Tables. Instead of Joinpoint results in the Figure 1, I recommend indicate the specific values of the incidence, mortality, and DALYs let say each three years of the period between 1992 and 2021 on the lines separately for male, female and all population. You also need indicate in what values are presented in the parts A,B, and C of the Figure one.

My recommendation is minor revision.

**Do you want your identity to be public for this peer review?** For information about this choice, including consent withdrawal, please see our Privacy Policy

Reviewer #1: No

Reviewer #2: No

---

## [Author Response · Author response to Decision Letter 1]

11 Dec 2025

We sincerely thank you for your thoughtful and constructive comments on our manuscript. These insights have been invaluable in helping us improve the quality and clarity of our work. We have carefully addressed all the points raised, and a point-by-point response is provided below. All changes in the manuscript have been highlighted in red for your convenience.

Reviewer #1:

1)There is a lack of detail on how data quality was assessed. Explicit information on how GBD data for china were validated should be added to the manuscript.

Response:We thank the reviewer for this important suggestion. The GBD study complies with the Guidelines for Accurate and Transparent Health Estimation Reporting. The prevalence and incidence data of Liver Cancer Attributable to Alcohol Use in China in the GBD database are mainly derived from national authoritative databases (e.g., the National Center for Disease Control and Prevention, the National Cancer Center and the National Bureau of Statistics of China). Currently, there have been tens of thousands of studies worldwide that have used the GBD database for analysis, and many of them are from China. These studies have been used to guide China's public health policies, thus verifying the validity and reliability of the database. We have now added a detailed description of the GBD data for China in the "Data collection" subsection within the "Methods"section.

2) Were multiple comparisons controlled for in the analyse? Is there the potential for inflated Type-I error? Either this statistical control should be used, or an explanation on why it is not needed should be provided.

Response:We appreciate the reviewer's keen methodological point. The Joinpoint software currently uses a series of permutation tests to determine the number of connection points. Before version 3.0, the software used Bonferroni adjustments to control the error probability of each multiitem test. The Bonferroni adjustment is relatively conservative because the actual overall significance level is usually lower than the nominal level α. The permutation test controls the overall probability of overfitting error and is superior to the traditional Bonferroni adjustment. Therefore, the permutation test used in this study significantly controls the probability of making type 1 errors.We have now explicitly addressed this in the "Statistical Analysis" section.

3)The project determined that the incidence of DALYs declines after age 65. However, could this reflect mortality levels (i.e., individuals with liver disease died at earlier ages) rather than a true reduction in DALYs? In this case, I believe this data would be censored and may need to be accounted for. This should be discussed as a potential limitation or controlled for.

Response:We fully acknowledge that the interpretation of the declining DALYs trend in the elderly population (≥65 years) may be influenced by this bias. Specifically, individuals who died from alcohol-attributable liver cancer before reaching the older age groups are not represented in the DALYs calculations for those groups, potentially leading to an underestimation of the true burden in the elderly and an overestimation of the magnitude of the decline. As correctly noted by the reviewer, and due to the nature of our secondary data analysis, we are unable to directly observe or statistically adjust for this effect at the individual level within the current GBD framework. Therefore, we have taken great care to discuss this limitation transparently in the revised manuscript (please see the 'Discussion'' sections), framing our conclusions with appropriate caution. We note that advanced statistical methods such as the Fine-Gray competing risk model or multi-state modeling with individual-level time-to-event data, could potentially help quantify and adjust for this bias. Addressing this complex issue with more granular data constitutes a valuable and clear direction for our future research.

4)The figures are difficult to see as they are very blurry. Please provide more appropriate figures for interpretation.

a. Further, it appears that no confidence intervals are included on Figure 1. It is difficulty to see.

Response:We sincerely apologize for the suboptimal quality of the figures in our initial submission.The new figure now displays the raw trends for incidence, DALYs, and mortality. Data points at regular intervals (every 3-5 years) have been marked and labeled with their specific values for males, females, and both sexes combined.

Reviewer #2:

To my opinion, the manuscript is well-written and demonstrates the interesting results for researchers working in similar area as authors. But still I have some recommendations how to improve the quality of the manuscript. To my opinion results presented in Figure 1 partially duplicates the results from Tables 1-3. So I recommend remove from the Figure 1 results of Joinpoint analysis which are already presented in the mentioned Tables. Instead of Joinpoint results in the Figure 1, I recommend indicate the specific values of the incidence, mortality, and DALYs let say each three years of the period between 1992 and 2021 on the lines separately for male, female and all population. You also need indicate in what values are presented in the parts A,B, and C of the Figure one.

Response:We thank the reviewer for the positive feedback and the excellent suggestion to improve the clarity and information value of Figure 1.

We have revised Figure 1 accordingly. The joinpoint segments have been removed. The new figure now displays the raw trends for incidence, DALYs, and mortality. Data points at regular intervals (every 3-5 years) have been marked and labeled with their specific values for males, females, and both sexes combined. The y-axis labels now clearly state that rates are presented as "per 100,000 population".

---

## [Decision Letter · Decision Letter 1]

2 Feb 2026

Trend of Liver Cancer Attributable to Alcohol Use in China from 1992 to 2021: An age-period-cohort analysis study

PONE-D-25-33288R1

Dear Dr. Dan Jia,

We’re pleased to inform you that your manuscript has been judged scientifically suitable for publication and will be formally accepted for publication once it meets all outstanding technical requirements.

Kind regards,

Ricardas Radisauskas

Academic Editor

PLOS One

Additional Editor Comments (optional):

Dear manuscript authors,

Thank you for your submitted manuscript.

I want to inform you that after revisions to the manuscript, your submitted article has been accepted for publication in this journal.

Reviewers' comments:

Reviewer's Responses to Questions

**Comments to the Author**

Reviewer #1: All comments have been addressed

Reviewer #2: All comments have been addressed

2. Is the manuscript technically sound, and do the data support the conclusions?

Reviewer #1: (No Response)

Reviewer #2: Yes

3. Has the statistical analysis been performed appropriately and rigorously?

Reviewer #1: (No Response)

Reviewer #2: Yes

4. Have the authors made all data underlying the findings in their manuscript fully available?

Reviewer #1: (No Response)

Reviewer #2: Yes

5. Is the manuscript presented in an intelligible fashion and written in standard English?

Reviewer #1: (No Response)

Reviewer #2: Yes

Reviewer #1: (No Response)

Reviewer #2: The authors of the manuscript modified the manuscript according to all my recommendations. My final recommendation is accept the manuscript for publication.

**Do you want your identity to be public for this peer review?** For information about this choice, including consent withdrawal, please see our Privacy Policy

Reviewer #1: No

Reviewer #2: **Yes:** Abdonas Tamosiunas

---

## [Editor Report · Acceptance letter]

PONE-D-25-33288R1

PLOS One

Dear Dr. Jia,

I'm pleased to inform you that your manuscript has been deemed suitable for publication in PLOS One. Congratulations! Your manuscript is now being handed over to our production team.

Kind regards,

on behalf of

Professor Ricardas Radisauskas

Academic Editor

PLOS One